# Grenade: Graph-Centric Language Model for Self-Supervised Representation Learning on Text-Attributed Graphs

**Yichuan Li**
Worcester Polytechnic Institute
yli29@wpi.edu

**Kaize Ding**[*]
Northwestern University
kaize.ding@northwestern.edu

**Kyumin Lee**
Worcester Polytechnic Institute
kmlee@wpi.edu

## Abstract

Self-supervised representation learning on text-attributed graphs, which aims to create expressive and generalizable representations for various downstream tasks, has received increasing research attention lately. However, existing methods either struggle to capture the full extent of structural context information or rely on task-specific training labels, which largely hampers their effectiveness and generalizability in practice. To solve the problem of self-supervised representation learning on text-attributed graphs, we develop a novel Graph-Centric Language model – Grenade. Specifically, Grenade exploits the synergistic effect of both pre-trained language model and graph neural network by optimizing with two specialized self-supervised learning algorithms: graph-centric contrastive learning and graph-centric knowledge alignment. The proposed graph-centric self-supervised learning algorithms effectively help Grenade to capture informative textual semantics as well as structural context information on text-attributed graphs. Through extensive experiments, Grenade shows its superiority over state-of-the-art methods. Implementation is available at https://github.com/bigheiniu/GRENADE.

## 1 Introduction

Text-Attributed Graph (TAG) (Yang et al., 2021) (*a.k.a.*, Textual Graph) has been widely used for modeling a variety of real-world applications, such as information retrieval (Cohan et al., 2020; Yang et al., 2021), product recommendation (Zhu et al., 2021) and many more. In TAG, each node represents a text document, while the relationships among these text nodes are depicted by the edges. For instance, in citation networks, text nodes represent academic papers, and edges are the citation relationship between different papers. To conduct different analytics tasks on TAG, the key is to learn expressive node representations for the text nodes.

Recent research has demonstrated that self-supervised learning (SSL) can substantially improve the effectiveness of representation learning on text data (Reimers and Gurevych, 2019; Gao et al., 2021; Wu et al., 2020) without using human supervision. Those methods are commonly learned under the assumption that text documents are independently and identically distributed (*i.i.d.*), which neglects the structural interdependencies among text nodes on TAG. However, the interdependencies between different text documents can provide valuable insights for understanding their semantic relationships. Take citation networks as an example, those academic papers (text nodes) that have citation relationships often share similar topics. Hence, it is necessary for SSL models to account for not only textual semantics but also structural context information.

In fact, self-supervised representation learning on TAG remains in its infancy: **(i)** Though recent research efforts (Zhao et al., 2022; Chien et al., 2021; Cohan et al., 2020; Yasunaga et al., 2022) try to empower pre-trained language models (PLM) with structural context information, most of them still stay superficial by designing local structure-dependent SSL objectives. For example, both GIANT (Chien et al., 2021) and SPECTER (Cohan et al., 2020) train the language model by inferring the local neighborhood based on representations of text nodes. However, simply relying on those SSL objectives cannot help the PLM fully understand complex graph structures, especially compared to models like graph neural networks (GNN) (Kipf and Welling, 2017; Velickovic et al., 2018; Hamilton et al., 2017; Ding et al., 2022a); **(ii)** Meanwhile, another line of research (Mavromatis et al., 2023; Zhao et al., 2022) try to combine the advantages of both PLM and GNN by distilling the knowledge from one to the other (Hinton et al., 2015) and have shown promising results. Nonetheless, one major issue is that those methods are task-specific (e.g.,

---

[*]Corresponding Author.

semi-supervised node classification) and require human-annotated labels to enable knowledge distillation. Such an inherent limitation jeopardizes the versatility of their models for handling different and even unseen downstream tasks, which runs counter to the goal of SSL.

To go beyond the existing learning paradigms and capture informative textual semantic and graph structure information, we develop a new model for self-supervised learning on TAG, namely GRENADE (Graph-Centric Language Model). GRENADE is built with a PLM encoder along with an adjuvant GNN encoder that provides complementary knowledge for it. More importantly, GRENADE is learned through two new self-supervised learning algorithms: Graph-Centric Contrastive Learning (GC-CL), a *structure-aware* and *augmentation-free* contrastive learning algorithm that improves the representation expressiveness by leveraging the inherent graph neighborhood information; and Graph-Centric Knowledge Alignment (GC-KA), which enables the PLM and GNN modules to reinforce each other by aligning their learned knowledge encoded in the text node representations.

Specifically, GC-CL enforces neighboring nodes to share similar semantics in the latent space by considering them as positive pairs. Even without using data augmentation, GC-CL performs node-wise contrastive learning to elicit the structural context information from TAG. In the meantime, GC-KA bridges the knowledge gap between PLM and GNN by performing dual-level knowledge alignment on the computed representations: at the node level, we minimize the distance between the representations learned from two encoders that focus on different modalities. At the neighborhood level, we minimize the discrepancy between two neighborhood similarity distributions computed from PLM and GNN. By virtue of the two proposed graph-centric self-supervised learning algorithms, we are able to learn GRENADE that can generate expressive and generalizable representations for various downstream tasks without using any human supervision. In summary, our work has the following contributions:

- We develop GRENADE, which is a graph-centric language model that addresses the underexplored problem of self-supervised learning on TAG.
- We propose two new self-supervised learning algorithms for TAG, which allow us to perform contrastive learning and knowledge alignment in a graph-centric way.

- We conduct extensive experiments to show that our model GRENADE significantly and consistently outperforms state-of-the-art methods on a wide spectrum of downstream tasks.

## 2 Problem Definition

**Notations.** We utilize bold lowercase letters such as $\mathbf{d}$ to represent vectors, bold capital letters like $\mathbf{W}$ to denote matrices and calligraphic capital letters like $\mathcal{W}$ to represent sets. Let $G = (\mathbf{A}, \mathcal{D})$ denote a text-attributed graph with adjacency matrix $\mathbf{A} \in \{0,1\}^{|D| \times |D|}$ and text set $\mathcal{D}$. The $\mathbf{A}_{ij} = 1$ when there is a connection between node $i$ and $j$. Each node $i$ represents a text document which consists of a sequence of *tokens* $\mathcal{D}_i = \{w_v\}_{v=0}^{|\mathcal{D}_i|}$.

**Problem 1** *Given an input text-attributed graph (*TAG*) denoted as $G=(\mathbf{A}, \mathcal{D})$, our goal is to learn a graph-centric language model* PLM$(\cdot)$, *that can generate expressive and generalizable representation for an arbitray node $i$ on $G$:* $\mathbf{d}_i=$PLM$(\mathcal{D}_i)$. *Note that the whole learning process is performed solely on the input graph $G$ without the utilization of human-annotated labels.*

## 3 Proposed Approach: Graph-Centric Language Model (GRENADE)

To learn expressive representations from TAG in a self-supervised learning manner, we propose our Graph-Centric Language Model GRENADE, which bridges the knowledge gap between Pre-trained Language Model (PLM) and Graph Neural Network (GNN). By optimizing two distinct encoders with a set of novel self-supervised learning algorithms, the PLM encoder and GNN encoder mutually reinforce each other, and we can finally derive our Graph-Centric Language Model (GRENADE). GNN. The overall framework is shown in Fig. 1.

### 3.1 Model Architecture

Our proposed model GRENADE is composed of a Pre-trained Language Model (PLM) along with a Graph Neural Network (GNN), which are optimized by a set of novel self-supervised learning algorithms. We first introduce the details about those two essential components as follows:

**PLM Encoder.** The primary component PLM$(\cdot)$ is a BERT (Devlin et al., 2018) based text encoder that projects a sequence of tokens $\mathcal{D}_i$ into a vectorized text node representation $\mathbf{d}_i$:

$$\mathbf{d}_i=\text{PLM}(\mathcal{D}_i), \qquad (1)$$

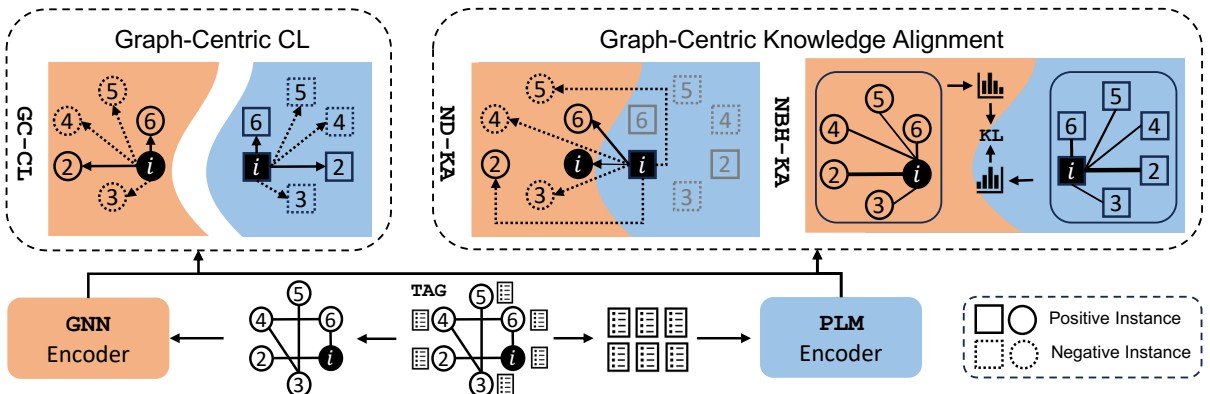

Figure 1: Illustration of GRENADE. Given a text-attributed graph (TAG), GRENADE is jointly optimized by two graph-centric self-supervised learning algorithms: graph-centric contrastive learning GC-CL and a dual-level graph-centric knowledge alignment, which comprises node-level alignment (ND-KA) and neighborhood-level alignment (NBH-KA).

where $\mathbf{d}_i$ is the hidden representation of the [CLS] token computed from the last layer of the PLM encoder.

**GNN Encoder.** As an adjuvant component, the GNN encoder $\text{GNN}(\cdot)$ is built with a stack of message-passing based GNN layers, which compute the node $i$'s representation by iteratively aggregating and transforming the feature information from its neighborhood (Hamilton et al., 2017). For each node $i$, its representation learned from a $L$-layer GNN encoder can be denoted as:

$$\mathbf{e}_i = \mathbf{E}_{[i,:]}, \mathbf{E} = \text{GNN}(\mathbf{E}^0, \mathbf{A}) \qquad (2)$$

where the input node feature matrix $\mathbf{E}^0$ is obtained from the hidden representations of [CLS] token from the last layer of a pre-trained BERT model.

### 3.2 Graph-Centric Contrastive Learning

In order to improve the learning capability of those two encoders without using any human-annotated labels, one prevailing way is to conduct contrastive learning from either the text perspective (Gao et al., 2021) or graph perspective (Ding et al., 2022c). However, most of the existing contrastive learning methods have the following two limitations: *(1)* conventional instance-level contrastive learning methods merely encourage instance-wise discrimination (Li et al., 2021b; Ding et al., 2023), which neglects the property of TAG, i.e., the relational information among text nodes. Hence, those instances that share similar semantics may be undesirably pushed away in the latent space; *(2)* existing methods commonly rely on arbitrary augmentation functions to generate different augmented views

for applying contrastive learning, while those augmentations may unexpectedly disturb the semantic meaning of the original instance (Lee et al., 2022).

To counter the aforementioned issues, we propose a new graph-centric contrastive learning (GC-CL) algorithm, which is *structure-aware* and *augmentation-free*. GC-CL exploits inherent graph knowledge from TAG and can be applied to both the PLM encoder and GNN encoder. As suggested by the Homophily principle (McPherson et al., 2001), neighboring nodes commonly share similar semantics, meaning that their representations should also be close to each other in the latent space. Based on the PLM representation of node $i$, its $K$-hop neighboring nodes $\mathcal{N}(i)$, and the node $i$ excluded mini-batch instances $\mathcal{B}(i)$, the GC-CL objective for PLM can be defined as follows:

$$\mathcal{L}_{\text{GC-CL}_1} = \frac{-1}{|\mathcal{N}(i)|} \sum_{p \in \mathcal{N}(i)} \log \frac{e^{\text{sim}(\mathbf{d}_i, \mathbf{d}_p)/\tau}}{\sum_{j \in \mathcal{C}(i)} e^{\text{sim}(\mathbf{d}_i, \mathbf{d}_j)/\tau}},$$
$$(3)$$

where $\tau$ denotes the temperature and $\text{sim}(\cdot, \cdot)$ represents the cosine similarity function. Here $\mathcal{C}(i) = \mathcal{N}(i) \cup \mathcal{B}(i)$. Note that for node $i$, we consider its PLM representation $\mathbf{d}_i$ as the query instance. The positive instances are the representations of node $i$'s $K$-hop neighboring nodes $\{\mathbf{d}_p | p \in \mathcal{N}(i)\}$. Meanwhile, the negative instances are the representations of other text nodes excluding $i$ within the same mini-batch $\{\mathbf{d}_j | j \in \mathcal{B}(i)\}$.

Similar to the PLM encoder, we also apply our GC-CL algorithm to the GNN encoder $\text{GNN}(\cdot)$. Specif-

ically, the objective function is defined as follows:

$$\mathcal{L}_{\text{GC-CL}_2} = \frac{-1}{|\mathcal{N}(i)|} \sum_{p \in \mathcal{N}(i)} \log \frac{e^{\text{sim}(\mathbf{e}_i, \mathbf{e}_p)/\tau}}{\sum_{j \in \mathcal{C}(i)} e^{\text{sim}(\mathbf{e}_i, \mathbf{e}_j)/\tau}},$$

(4)

where $\mathbf{e}_i$ is the query instance. The positive instances are $\{\mathbf{e}_p | p \in \mathcal{N}(i)\}$ and the negative instances are $\{\mathbf{e}_j | j \in \mathcal{B}(i)\}$.

Apart from the conventional instance-level contrastive learning counterparts, our graph-centric contrastive learning also enforces neighboring nodes to share similar representations. In a sense, this self-supervised learning algorithm is analogous to performing link prediction task based on the representations learned from the PLM encoder, which inherently elicits informative graph knowledge during the learning process.

### 3.3 Graph-Centric Knowledge Alignment

In this work, our ultimate goal is to learn expressive and generalizable representations that encode informative textual semantics within each text node as well as the relational information among nodes. However, individually conducting the graph-centric contrastive learning on either PLM or GNN is not enough due to the lack of knowledge exchange between them. To better align and enhance the knowledge captured by the PLM and GNN encoders, we propose a dual-level graph-centric knowledge alignment algorithm for TAG, which includes Node-Level Knowledge Alignment (ND-KA) and Neighborhood-Level Knowledge Alignment (NBH-KA).

**Node-Level Knowledge Alignment.** Different from the previously introduced graph-centric contrastive learning, which only focuses on single-modal contrasting, ND-KA tries to align the knowledge across the two encoders by performing graph-centric contrastive learning in a cross-modal form. For each node $i$, based on its representations learned from the PLM encoder and GNN encoder (i.e., $\mathbf{d}_i$ and $\mathbf{e}_i$, respectively), we formulate the objective of ND-KA as follows:

$$\mathcal{L}_{\text{ND-KA}} = \frac{-1}{|\widetilde{\mathcal{N}}(i)|} \sum_{p \in \widetilde{\mathcal{N}}(i)} \left( \log \frac{e^{\text{sim}(\mathbf{e}_i, \mathbf{d}_p)/\tau}}{\sum_{j \in \widetilde{\mathcal{C}}(i)} e^{\text{sim}(\mathbf{e}_i, \mathbf{d}_j)/\tau}} \right.$$
$$\left. + \log \frac{e^{\text{sim}(\mathbf{d}_i, \mathbf{e}_p)/\tau}}{\sum_{j \in \widetilde{\mathcal{C}}(i)} e^{\text{sim}(\mathbf{d}_i, \mathbf{e}_j)/\tau}} \right) / 2,$$

(5)

where $\widetilde{\mathcal{N}}(i) = \{i\} \cup \mathcal{N}(i)$ and $\tilde{\mathcal{C}}(i) = \widetilde{\mathcal{N}}(i) \cup \mathcal{B}(i)$. Note that for node $i$, we first consider $\mathbf{e}_i$

that is learned from the GNN encoder as the query, then construct the positive and negative instances based on the representations learned from the PLM encoder. Specifically, the positive instances include both the representation of node $i$ as well as the representations of $i$'s $K$-hop neighboring nodes (i.e., $\{\mathbf{d}_p | p \in \widetilde{\mathcal{N}}(i)\}$), and the negative instances are the representations of other instances within the same mini-batch $\{\mathbf{d}_j | j \in \mathcal{B}(i)\}$. In the meantime, we also consider the $\mathbf{d}_i$ as the query and construct its corresponding positive and negative instances in the same way. Here we omit the illustration for simplicity.

By virtue of the proposed ND-KA algorithm, the representations of the same node learned from two separate encoders will be pulled together in the latent space. In the meantime, ND-KA also encourages neighboring nodes to have similar representations across different modalities.

**Neighborhood-Level Knowledge Alignment.** To further facilitate knowledge alignment between PLM and GNN, we propose Neighborhood-Level Knowledge Alignment (NBH-KA) to align the neighborhood similarity distributions learned from the two encoders. Specifically, NBH-KA first computes the neighborhood similarity distribution between the query node $i$ and its $K$-hop neighboring nodes $\mathcal{N}(i)$ as well as the rest nodes within the same mini-batch $\mathcal{B}(i)$ for each encoder. Then we minimize the KL-divergence between the two distributions to align the knowledge between two encoders. The corresponding learning objective is:

$$\mathcal{L}_{\text{NBH-KA}} = \Big( \text{KL}(P_{\text{PLM}}(i) || P_{\gamma}(i))$$
$$+ \text{KL}(P_{\text{GNN}}(i) || P_{\text{PLM}}(i)) \Big) / 2,$$
$$P_{\text{PLM}}(i) = \text{softmax}_{j \in \mathcal{C}(i)}(\text{sim}(\mathbf{d}_i, \mathbf{d}_j)/\tau),$$
$$P_{\text{GNN}}(i) = \text{softmax}_{j \in \mathcal{C}(i)}(\text{sim}(\mathbf{e}_i, \mathbf{e}_j)/\tau),$$

(6)

where $P_{\text{PLM}}(i)$ and $P_{\text{GNN}}(i)$ are the neighborhood similarity distributions for PLM encoder and GNN encoder respectively. From a certain perspective, our NBH-KA algorithm can be considered a self-supervised form of knowledge distillation. Specifically, NBH-KA leverages the neighborhood information as self-supervision to guide the knowledge alignment process. Moreover, we conduct two-way knowledge alignment across two encoders, which is different from original knowledge distillation.

## 3.4 Model Learning

In order to learn our graph-centric language model GRENADE on TAG without using human-annotated labels, we jointly optimize the proposed graph-centric contrastive learning and knowledge alignment algorithms. For the sake of simplicity, we define the overall training loss as follows:

$$\mathcal{L} = \mathcal{L}_{\text{GC-CL}_1} + \mathcal{L}_{\text{GC-CL}_2} + \mathcal{L}_{\text{ND-KA}} + \mathcal{L}_{\text{NBH-KA}}. \quad (7)$$

Once the training is finished, we can freeze the parameters of the PLM encoder and use it to compute the representations of each text node with a forward pass. The computed representations can be further used for different downstream tasks.

## 4 Experiment

To evaluate the effectiveness of our approach GRENADE, we conduct comprehensive experiments on different datasets and various downstream tasks.

### 4.1 Experimental Setup

**Evaluation Datasets.** We evaluate the generalizability of the representations computed from different methods on three Open Graph Benchmark (OGB) (Hu et al., 2020) datasets: ogbn-arxiv, ogbn-products, ogbl-citation2. These datasets are utilized to evaluate the performance of few-shot and full-shot node classification, node clustering, and link prediction tasks. It should be noted that ogbn-arxiv and ogbn-products datasets are not originally designed for link prediction evaluation. Therefore, we create two link prediction tasks based on these two datasets, respectively. Furthermore, we incorporate obgl-citation2 into our node classification experiment. The statistical information of the datasets is shown in Tab. 1. More comprehensive information regarding the dataset extension can be found in Appendix A.

| Dataset | #Nodes | #Edges | #Classes |
|---|---|---|---|
| ogbn-arxiv | 169,343 | 1,166,243 | 40 |
| ogbn-products | 2,449,029 | 61,859,140 | 47 |
| ogbl-citations2 | 2,927,963 | 30,387,995 | 172 |

Table 1: Statistical information of the datasets.

**Baseline Methods.** The baseline methods included in our study encompass three categories: *(1) Untuned representations:* BERT (Devlin et al., 2018) and OGB representations (Mikolov et al., 2013; Hu et al., 2020). For BERT, we extract the final

layer's hidden state of [CLS] token from frozen bert-base-uncased as the text node representation. As for OGB, we utilize the default features from benchmark datasets, such as averaged word embeddings and bag-of-words representations. *(2) Text-only self-supervised representation learning models:* BERT+MLM and SimCSE (Gao et al., 2021); In BERT+MLM, we apply masked language modeling to BERT for the target TAG. SimCSE employs instance-wise contrastive learning to learn text node representation. *(3) Structure-augmented language models:* This category includes SPECTER (Cohan et al., 2020), GIANT (Chien et al., 2021) and GLEM (Zhao et al., 2022). SPECTER applies graph-centric contrastive learning on the language model, and GIANT employs the extreme multi-label classification to train the language model for neighborhood prediction. It is noteworthy that GLEM utilizes task-specific labels to alternatively guide the pre-trained language model PLM and graph neural networks GNN through self-knowledge distillation. Compared with GLEM, our proposed method GRENADE is fully self-supervised and does not rely on any human-annotated labels. The learned text node representations can be efficiently and effectively generalized to downstream tasks.

**Implementation Details.** To ensure a fair comparison, we implemented all baseline methods and GRENADE using the same language model, specifically bert-base-uncased. For our proposed method, GRENADE, we set the $K$-hop neighbor as 1, set the temperature parameter $\tau$ to 0.05 in all the loss functions. The optimal hyperparameter $|\mathcal{N}(i)|$ is discussed in § 4.5. Please refer to Appendix B for additional implementation details.

### 4.2 Experimental Results

**Few-shot Node Classification.** To assess the generalizability of learned representation to new tasks under low-data scenarios, we conduct experiments on few-shot node classification. Under this setting, the classification models are trained with varying numbers of labeled instances per class ($k = \{2, 4, 8, 16\}$). We repeat the experiment 10 times and reported the average results along with the standard deviation. The classification models utilized in this evaluation are the multilayer perceptron (MLP) and GraphSAGE (Hamilton et al., 2017). The hyperparameters for the classification models can be found in Appendix B. As the result showed in Tab. 2, several observations can be made: *(1)* In most cases,

| Methods | MLP | | | | GraphSAGE | | | |
|---|---|---|---|---|---|---|---|---|
| | $k=2$ | $k=4$ | $k=8$ | $k=16$ | $k=2$ | $k=4$ | $k=8$ | $k=16$ |
| ogbn-arxiv | | | | | | | | |
| OGB$^\star$ | $32.16_{\pm1.96}$ | $37.81_{\pm1.62}$ | $42.33_{\pm1.07}$ | $45.84_{\pm0.47}$ | $49.91_{\pm3.46}$ | $55.52_{\pm1.42}$ | $59.30_{\pm1.14}$ | $62.21_{\pm0.58}$ |
| BERT | $34.06_{\pm2.73}$ | $40.29_{\pm2.16}$ | $46.59_{\pm0.88}$ | $50.17_{\pm1.02}$ | $52.92_{\pm3.21}$ | $57.11_{\pm1.06}$ | $60.36_{\pm1.17}$ | $64.10_{\pm0.61}$ |
| BERT+MLM | $38.41_{\pm2.11}$ | $46.72_{\pm1.72}$ | $51.89_{\pm0.98}$ | $55.87_{\pm1.23}$ | $53.02_{\pm1.26}$ | $57.76_{\pm1.41}$ | $61.94_{\pm0.77}$ | $65.22_{\pm0.49}$ |
| SimCSE | $28.83_{\pm1.68}$ | $32.65_{\pm1.46}$ | $37.78_{\pm1.26}$ | $43.25_{\pm0.69}$ | $46.61_{\pm2.97}$ | $53.86_{\pm1.20}$ | $57.75_{\pm0.89}$ | $62.39_{\pm0.61}$ |
| SPECTER | $50.15_{\pm2.21}$ | $54.46_{\pm0.96}$ | $58.74_{\pm0.63}$ | $61.63_{\pm0.78}$ | $53.85_{\pm2.27}$ | $59.46_{\pm1.63}$ | $63.43_{\pm0.61}$ | $66.41_{\pm0.45}$ |
| GIANT$^\star$ | $48.50_{\pm2.30}$ | $55.72_{\pm1.90}$ | $59.80_{\pm1.03}$ | $64.15_{\pm0.87}$ | $50.18_{\pm2.46}$ | $55.30_{\pm1.69}$ | $59.24_{\pm1.33}$ | $63.48_{\pm0.77}$ |
| GLEM | - | - | - | - | $27.14_{\pm2.31}$ | $45.52_{\pm1.27}$ | $53.37_{\pm1.08}$ | $55.39_{\pm0.89}$ |
| GRENADE | $\mathbf{55.85}_{\pm2.34}$ | $\mathbf{61.10}_{\pm1.59}$ | $\mathbf{63.95}_{\pm0.89}$ | $\mathbf{66.62}_{\pm0.47}$ | $\mathbf{57.17}_{\pm3.54}$ | $\mathbf{60.49}_{\pm0.93}$ | $\mathbf{64.65}_{\pm1.08}$ | $\mathbf{67.50}_{\pm0.41}$ |
| ogbn-products | | | | | | | | |
| OGB$^\star$ | $9.47_{\pm1.51}$ | $13.54_{\pm1.42}$ | $16.83_{\pm2.63}$ | $20.71_{\pm1.32}$ | $24.74_{\pm3.01}$ | $33.14_{\pm2.58}$ | $38.38_{\pm1.18}$ | $44.19_{\pm2.61}$ |
| BERT | $20.53_{\pm4.03}$ | $30.91_{\pm2.64}$ | $40.82_{\pm1.52}$ | $47.77_{\pm1.07}$ | $40.53_{\pm2.87}$ | $50.43_{\pm1.73}$ | $57.29_{\pm1.48}$ | $59.03_{\pm0.48}$ |
| BERT+MLM | $38.54_{\pm3.35}$ | $47.15_{\pm3.41}$ | $55.95_{\pm0.89}$ | $59.57_{\pm1.53}$ | $54.73_{\pm4.35}$ | $60.80_{\pm2.65}$ | $64.63_{\pm1.94}$ | $67.75_{\pm1.48}$ |
| SimCSE | $7.76_{\pm1.22}$ | $11.30_{\pm1.30}$ | $17.94_{\pm1.91}$ | $25.61_{\pm0.72}$ | $24.99_{\pm5.01}$ | $37.05_{\pm1.73}$ | $44.74_{\pm1.96}$ | $50.13_{\pm2.94}$ |
| SPECTER | $27.86_{\pm4.14}$ | $43.70_{\pm2.70}$ | $51.51_{\pm1.54}$ | $57.63_{\pm1.45}$ | $42.74_{\pm4.58}$ | $51.91_{\pm2.31}$ | $57.72_{\pm2.52}$ | $60.77_{\pm1.36}$ |
| GIANT$^\star$ | $20.83_{\pm2.68}$ | $31.37_{\pm1.95}$ | $42.84_{\pm2.72}$ | $51.36_{\pm2.19}$ | $30.90_{\pm4.63}$ | $40.80_{\pm4.01}$ | $52.34_{\pm2.20}$ | $58.58_{\pm2.36}$ |
| GLEM | - | - | - | - | $41.72_{\pm4.62}$ | $42.50_{\pm3.45}$ | $43.05_{\pm2.78}$ | $48.64_{\pm2.01}$ |
| GRENADE | $\mathbf{38.60}_{\pm4.51}$ | $\mathbf{49.64}_{\pm1.39}$ | $\mathbf{59.34}_{\pm2.38}$ | $\mathbf{65.05}_{\pm1.00}$ | $\mathbf{59.63}_{\pm2.80}$ | $\mathbf{64.95}_{\pm1.63}$ | $\mathbf{67.38}_{\pm2.17}$ | $\mathbf{70.92}_{\pm1.18}$ |
| ogbl-citation2 | | | | | | | | |
| OGB$^\star$ | $21.78_{\pm1.55}$ | $25.54_{\pm1.55}$ | $27.39_{\pm0.69}$ | $29.53_{\pm0.70}$ | $18.83_{\pm3.12}$ | $22.49_{\pm1.84}$ | $31.38_{\pm1.89}$ | $35.19_{\pm0.89}$ |
| BERT | $17.95_{\pm2.34}$ | $20.84_{\pm2.17}$ | $23.84_{\pm1.36}$ | $26.77_{\pm1.38}$ | $21.05_{\pm2.49}$ | $25.65_{\pm1.58}$ | $28.06_{\pm2.03}$ | $35.84_{\pm1.48}$ |
| BERT+MLM | $32.03_{\pm1.84}$ | $34.39_{\pm2.61}$ | $38.53_{\pm1.82}$ | $41.66_{\pm0.98}$ | $26.28_{\pm2.35}$ | $28.10_{\pm2.47}$ | $37.69_{\pm1.39}$ | $42.63_{\pm1.21}$ |
| SimCSE | $13.27_{\pm1.08}$ | $16.68_{\pm0.62}$ | $20.01_{\pm1.03}$ | $23.57_{\pm0.77}$ | $20.01_{\pm2.04}$ | $28.11_{\pm2.90}$ | $28.17_{\pm1.06}$ | $31.71_{\pm1.28}$ |
| SPECTER | $30.81_{\pm2.94}$ | $35.31_{\pm1.92}$ | $39.74_{\pm1.33}$ | $42.31_{\pm0.64}$ | $31.57_{\pm2.04}$ | $35.66_{\pm2.18}$ | $37.23_{\pm2.70}$ | $45.59_{\pm1.82}$ |
| GIANT$^\star$ | $38.93_{\pm1.81}$ | $43.74_{\pm1.93}$ | $48.81_{\pm1.25}$ | $52.05_{\pm0.72}$ | $35.75_{\pm2.72}$ | $38.43_{\pm3.43}$ | $40.99_{\pm3.42}$ | $49.44_{\pm1.68}$ |
| GLEM | - | - | - | - | $30.86_{\pm4.62}$ | $33.78_{\pm1.88}$ | $47.36_{\pm2.73}$ | $51.42_{\pm1.41}$ |
| GRENADE | $\mathbf{46.40}_{\pm2.00}$ | $\mathbf{47.93}_{\pm1.34}$ | $\mathbf{50.61}_{\pm0.71}$ | $\mathbf{53.75}_{\pm0.82}$ | $\mathbf{40.63}_{\pm3.77}$ | $\mathbf{44.44}_{\pm2.63}$ | $\mathbf{49.15}_{\pm1.73}$ | $\mathbf{52.41}_{\pm1.94}$ |

Table 2: Experiment results of few-shot node classification. $^\star$ indicates that the text node representations are obtained from their official release. $-$ indicates no result for GLEM. This is because in the representation learning stage, GLEM will utilize the labeled dataset to train GNN.

| Methods | ogbn-arxiv | | | ogbn-products | | | ogbl-citation2 | | |
|---|---|---|---|---|---|---|---|---|---|
| | ACC | ARI | NMI | ACC | ARI | NMI | ACC | ARI | NMI |
| OGB | $39.57_{\pm0.39}$ | $12.76_{\pm1.22}$ | $17.43_{\pm0.79}$ | $38.53_{\pm1.43}$ | $16.46_{\pm4.24}$ | $20.21_{\pm1.84}$ | $40.31_{\pm0.40}$ | $22.82_{\pm0.34}$ | $40.57_{\pm0.38}$ |
| BERT | $35.65_{\pm1.21}$ | $8.41_{\pm1.01}$ | $12.78_{\pm1.20}$ | $48.72_{\pm0.57}$ | $52.17_{\pm1.81}$ | $35.76_{\pm0.82}$ | $40.57_{\pm0.31}$ | $22.58_{\pm0.84}$ | $33.38_{\pm0.44}$ |
| BERT+MLM | $40.07_{\pm0.60}$ | $15.05_{\pm0.17}$ | $19.24_{\pm0.51}$ | $64.35_{\pm0.81}$ | $69.58_{\pm1.31}$ | $54.64_{\pm0.31}$ | $49.13_{\pm0.46}$ | $31.91_{\pm0.46}$ | $44.44_{\pm0.34}$ |
| SimCSE | $33.42_{\pm0.74}$ | $5.80_{\pm0.40}$ | $9.62_{\pm0.74}$ | $39.40_{\pm1.02}$ | $29.00_{\pm2.22}$ | $19.88_{\pm1.01}$ | $26.67_{\pm0.64}$ | $8.49_{\pm0.41}$ | $14.26_{\pm0.65}$ |
| SPECTER | $58.00_{\pm0.66}$ | $40.44_{\pm1.17}$ | $42.81_{\pm0.53}$ | $70.15_{\pm0.67}$ | $\mathbf{69.82}_{\pm1.25}$ | $58.99_{\pm0.82}$ | $63.36_{\pm0.36}$ | $48.66_{\pm0.14}$ | $58.96_{\pm0.23}$ |
| GIANT | $58.00_{\pm0.82}$ | $39.69_{\pm1.22}$ | $43.73_{\pm0.68}$ | $61.99_{\pm0.78}$ | $47.60_{\pm4.18}$ | $47.51_{\pm1.14}$ | $63.06_{\pm0.53}$ | $48.57_{\pm0.89}$ | $58.68_{\pm0.25}$ |
| GRENADE | $\mathbf{61.96}_{\pm0.79}$ | $\mathbf{44.98}_{\pm1.85}$ | $\mathbf{49.19}_{\pm0.63}$ | $\mathbf{73.54}_{\pm0.75}$ | $69.64_{\pm1.64}$ | $\mathbf{64.14}_{\pm0.71}$ | $\mathbf{64.89}_{\pm0.30}$ | $\mathbf{50.22}_{\pm0.42}$ | $\mathbf{59.68}_{\pm0.23}$ |

Table 3: Experiment results of node clustering.

SSL based methods achieve better performance than non-SSL methods (BERT+MLM,SPECTER and GIANT > GLEM), this indicates the significance of SSL in enhancing model transferability to new tasks with limited labels. *(2)* Among state-of-the-art TAG representation models, GRENADE achieves the best performance on these datasets. This indicates the superior generalization ability of representations extracted by GRENADE. The designed knowledge alignment allows the GRENADE to integrate the pre-trained knowledge from PLM encoder and structure inductive bias learned by GNN encoder. These expressive representations can be easily and efficiently generalized to few-shot learning tasks.

**Full Data Node Classification.** We also conduct the node classification experiment with full training dataset under MLP, GraphSAGE (Hamilton et al., 2017) and RevGAT-KD (Li et al., 2021a). As the result shown in Tab. 4, we can observe that: *(1)* GRENADE achieves the best performance across all the baseline methods. *(2)* The performance gap between GRENADE and some baseline methods like GIANT and GLEM becomes smaller as more labeled data provided, but GRENADE is consistently better than these methods.

**Node Clustering.** In the node clustering task, we utilize the learned text node representations to train a K-means++ model for clustering instances. We apply the default hyperparameters of K-means++ as provided by scikit-learn (Pedregosa et al., 2011). The number of clusters is set to the number of classes in the dataset, and we assign the cluster label

| Methods | arxiv | | | products | | citation2 | |
|---|---|---|---|---|---|---|---|
| | MLP | GraphSAGE | RevGAT-KD | MLP | GraphSAGE | MLP | GraphSAGE |
| OGB | $55.50_{\pm0.23}{}^{\star}$ | $71.49_{\pm0.27}{}^{\star}$ | $74.26_{\pm0.17}{}^{\star}$ | $61.06_{\pm0.08}{}^{\star}$ | $75.81_{\pm0.46}$ | $48.98_{\pm1.74}$ | $62.10_{\pm0.25}$ |
| BERT | $62.91_{\pm0.60}{}^{\star}$ | $70.97_{\pm0.33}{}^{\star}$ | $73.59_{\pm0.10}{}^{\star}$ | $60.90_{\pm1.09}{}^{\star}$ | $80.70_{\pm0.50}$ | $56.75_{\pm0.33}$ | $60.38_{\pm0.67}$ |
| BERT+MLM | $67.71_{\pm0.22}$ | $73.65_{\pm0.14}$ | $75.39_{\pm0.16}$ | $76.86_{\pm0.24}$ | $80.90_{\pm0.11}$ | $66.11_{\pm0.35}$ | $64.57_{\pm0.50}$ |
| SimCSE | $60.58_{\pm0.13}$ | $67.26_{\pm0.24}$ | $73.69_{\pm0.15}$ | $67.69_{\pm0.06}$ | $80.41_{\pm0.44}$ | $53.91_{\pm0.36}$ | $58.75_{\pm0.24}$ |
| SPECTER | $70.40_{\pm0.25}$ | $74.19_{\pm0.18}$ | $75.61_{\pm0.67}$ | $75.38_{\pm0.22}$ | $79.42_{\pm0.34}$ | $54.20_{\pm0.67}$ | $66.58_{\pm0.23}$ |
| GIANT | $73.08_{\pm0.06}{}^{\star}$ | $74.59_{\pm0.28}{}^{\star}$ | $76.12_{\pm0.16}{}^{\star}$ | $79.82_{\pm0.07}{}^{\star}$ | $82.03_{\pm0.65}$ | $67.24_{\pm0.27}$ | $70.32_{\pm0.27}$ |
| GLEM | - | $73.59_{\pm0.40}$ | - | - | $82.02_{\pm0.62}$ | - | $68.25_{\pm0.18}$ |
| Grenade | $\mathbf{73.16}_{\pm0.12}$ | $\mathbf{75.00}_{\pm0.19}$ | $\mathbf{76.21}_{\pm0.17}$ | $\mathbf{81.58}_{\pm0.18}$ | $\mathbf{83.11}_{\pm0.56}$ | $\mathbf{68.11}_{\pm0.34}$ | $\mathbf{70.89}_{\pm0.34}$ |

Table 4: Supervised node classification performance comparison on benchmark datasets. Boldfaced numbers indicate the best performance of downstream models. The $\star$ represents the experiment results adopted from (Chien et al., 2021), while $\dagger$ denotes the experiment results adopted from (Zhao et al., 2022).

based on the most common label within each cluster. Following the evaluation protocol described in (Ding et al., 2022b), we report three clustering evaluation metrics: accuracy (ACC), normalized mutual information (NMI), and adjusted rand index (ARI). We exclude the GLEM model from this evaluation since it requires the labels during representation learning. To ensure robustness, we perform 10 runs of K-means++ with different random seeds and report the average results. As shown in Table 3, we observe that the structure augmented SSL methods outperform the text-only self-supervised representation learning methods (Grenade, GIANT, SPECTER > BERT+MLM, SimCSE). This indicates structure-augmented SSL methods can understand the context within graph structure that can lead to more accurate node representations, which in turn can lead to better clustering. Additionally, our proposed method Grenade consistently outperforms all baseline methods. The improvement demonstrates that Grenade can better preserve neighborhood information which will inform the clustering methods of how data points are interconnected or related to each other.

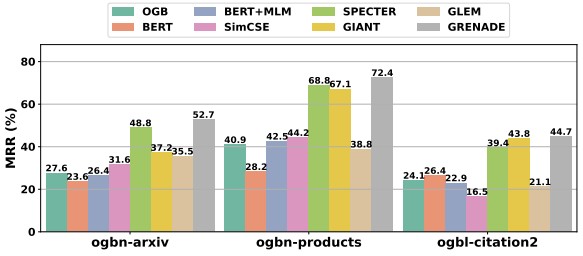

Figure 2: Experiment results of link prediction.

**Link Prediction.** Next, we evaluate the learned representation in predicting missing connections given existing connections from TAG. We aim to rank the positive candidates (1 or 2 positive instances) higher than the negative candidates (1,000 negative instances) for each query node. The eval-

uation metric used for this task is the mean reciprocal rank (MRR), which measures the reciprocal rank of the positive instance among the negative instances for each query instance and takes the average over all query instances. As shown in Fig. 2, we observe that Grenade significantly outperforms other approaches. In fact, Grenade achieves at least a 4% performance improvement compared to methods that utilize structure-augmented self-supervised learning loss (SPECTER and GIANT) across all datasets. This demonstrates that Grenade can better preserve the neighborhood information, which is consistent with the findings from § 4.2.

### 4.3 Representation Visualization

To visually demonstrate the quality of the learned representations, we apply t-distributed stochastic neighbor embedding (t-SNE) (Van der Maaten and Hinton, 2008) to for representation visualization. We compare Grenade with two best-performing baseline methods, including SPECTER and GIANT on the arxiv dataset. In Fig. 3, we present the t-SNE visualization of the embeddings for 10 randomly sampled classes comprising 5,000 subsampled instances. The colors in the visualization correspond to the labels of these subsampled instances. From Fig. 3, we observe Grenade exhibits denser clusters and more explicit boundaries among different classes compared to the baseline methods. This observation confirms that Grenade can learn compact intra-class and distinct inter-class representations.

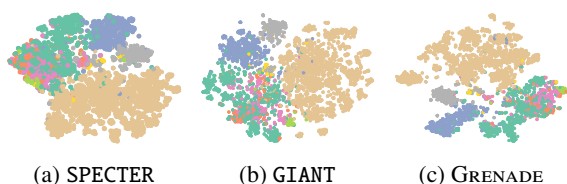

| (a) SPECTER | (b) GIANT | (c) Grenade |
|---|---|---|

Figure 3: Representation visualization on arxiv dataset.

## 4.4 Ablation Study

To validate the effectiveness of graph-centric contrastive learning and graph-centric knowledge alignment, we conducted an ablation study on GRENADE. In this study, we respectively remove GC-CL, ND-KA, and NBH-KA from the full model and report these model variants' performance in Tab. 5. In general, the full model GRENADE has the best performance in most cases, and we notice a performance decline when any of the components is removed or replaced, underscoring the significance of each component in GRENADE. Remarkably, we observe a performance improvement in link prediction after removing graph-centric contrastive learning (w/o GC-CL > GRENADE in terms of MRR). Considering the task similarity between GC-CL and the link prediction, one possible explanation is that removing GC-CL could help the model mitigate overfitting and further improve performance for the link prediction task. Meanwhile, this observation, in turn shows that the dual-level graph-centric knowledge alignment (ND-KA and NBH-KA) is effective for capturing structural context information from the TAG.

| Methods | MLP(ACC) | K-means++(ACC) | MRR | Avg. Rank↓ |
|---------|----------|----------------|-----|------------|
| GRENADE | $73.16_{\pm0.12}$ | $61.96_{\pm0.79}$ | 52.73 | **1.33** |
| w/o GC-CL | $72.83_{\pm0.08}$ | $58.75_{\pm0.99}$ | **55.87** | 2.67 |
| w/o ND-KA | $72.65_{\pm0.16}$ | $60.69_{\pm0.72}$ | 49.15 | 3.33 |
| w/o NBH-KA | $73.05_{\pm0.17}$ | $60.50_{\pm0.86}$ | 52.56 | 2.67 |

Table 5: Ablation study of graph-centric contrastive learning (GC-CL) and knowledge-alignment on ogbn-arxiv datasets. "w/o" is the abbreviation of "without".

## 4.5 Hyperparameter Analysis

$K$-hop Neighbors. We delved into understanding the impact of the $K$-hop neighbor selection on GRENADE's efficiency. The choice of different $K$ values directly affects the formulation of positive pairs in graph-centric contrastive learning (Eq. 3 and Eq. 4), and the alignment of knowledge between the graph neural network and the language model (Eq. 5 and Eq. 6). Based on the results presented in Fig. 4, it is evident that augmenting the hop distance adversely affects performance metrics in full data node classification (ACC of MLP), node clustering (ACC), and link prediction (MRR). This suggests that 1-hop neighbors optimally capture structural knowledge within our algorithm. However, when extending to 2-hop or 3-hop neighbors, there's a heightened risk of integrating noisy data. This insight aligns with the conclusions drawn from related research, specifically SPECTER (Cohan et al., 2020).

We contend that our methodology strikes a harmonious balance between assimilating structural data and filtering out extraneous noise, thereby ensuring consistent performance in our assessments.

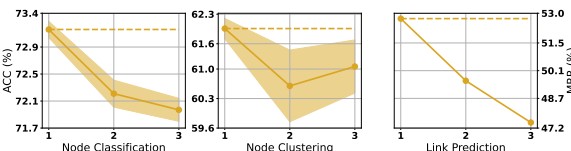

Figure 4: Hyperparameter evaluation for $K$-hop neighbors in the ogbn-arxiv dataset. Dashed lines indicate the peak performance of GRENADE with $K = 1$.

$1$-Hop Neighbor Size. One crucial aspect of GRENADE's SSL objectives is the hyperparameter $|\mathcal{N}(i)|$, which controls the number of 1-hop neighbors considered for representation learning. To investigate the impact of subsampled neighbor size in GRENADE, we conduct a hyperparameter analysis on the full training dataset node classification, node clustering, and link prediction tasks. As shown in Fig. 5, we observe that GRENADE achieves its best performance with a practical number of neighbors ($|\mathcal{N}(i)| = 2$ for ogbn-arxiv and $|\mathcal{N}(i)| = 1$ for ogbn-products). This finding is particularly advantageous as it reduces the computational burden of the PLM encoder in graph-centric contrastive learning and knowledge alignment between the PLM and GNN encoders.

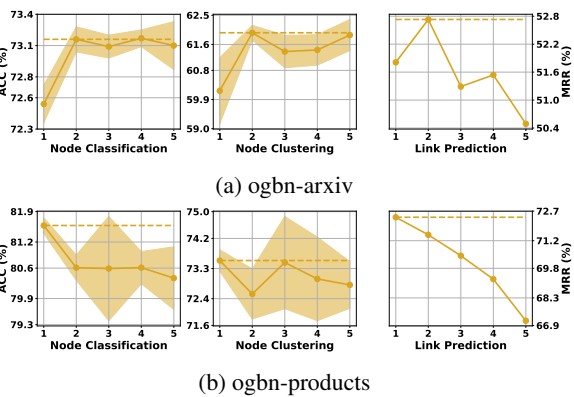

(a) ogbn-arxiv

(b) ogbn-products

Figure 5: Analysis of hyperparameters on both ogbn-arxiv and ogbn-products datasets. The dashed line represents GRENADE's optimal result when $|\mathcal{N}(i)| = 2$ for ogbn-arxiv and $|\mathcal{N}(i)| = 1$ for ogbn-products.

## 5 Related Work

Learning with TAG. This problem involves learning text node representations that encode both textual semantics and structural context information.

A prominent approach uses structure-augmented SSL objectives to incorporate structure information into language models (Chien et al., 2021; Cohan et al., 2020; Yasunaga et al., 2022; Ostendorff et al., 2022). For example, SPECTER and LinkBERT focus on node-pair relationships (Cohan et al., 2020; Yasunaga et al., 2022), while GIANT targets extreme multiclass neighborhood prediction (Chien et al., 2021). A concurrent work PATTON (Jin et al., 2023) trains a language model through network-contextualized masked language modeling and link prediction. A parallel line of research tries to integrate pre-trained language models (PLMs) with graph neural networks (GNN) (Yang et al., 2021; Zhao et al., 2022; Mavromatis et al., 2023; Shang et al., 2020; Zhang et al., 2021). GraphFormers interweaves GNN with LMs' self-attention layers (Yang et al., 2021). GLEM (Zhao et al., 2022), GraDBERT (Mavromatis et al., 2023) and LRTN (Zhang et al., 2021) leverage labeled datasets for co-training between PLM and GNN. While they excel in node classification benchmarks, their reliance on labeled datasets limits the representation's generalization ability to new tasks.

**Contrastive Learning.** Contrastive learning is a self-supervised learning paradigm that aims to learn representations by distinguishing between positive and negative instances (Jaiswal et al., 2020). A key practice in contrastive learning is to use augmented versions of the same instance as positive instances and other instances as negative instances (Gao et al., 2021; He et al.; Radford et al., 2021). For example, SimCSE creates augmented views for each instance based on dropout (Srivastava et al., 2014). However, conventional instance-level contrastive learning only encourages instance-wise discrimination (Li et al., 2021b) and commonly assumes different instances are *i.i.d.*, which neglects the relationships among different instances on TAG. Hence, conventional contrastive learning methods are ineffective to learn expressive representation learning on TAG. To address those limitations, many recent methods seek to extend the design of positive pair construction by considering local neighborhood information (Cohan et al., 2020; Ostendorff et al., 2022). However, those methods cannot fully capture complex graph structures. In contrast, our proposed method, GRENADE, leverages graph-centric contrastive learning and graph-centric knowledge alignment to fully exploit the structural context information from TAG.

## 6 Conclusion

In this paper, we introduce a self-supervised graph-centric language model: GRENADE, for learning expressive and generalized representation from textual attributed graphs (TAG). GRENADE is learned through two self-supervised learning algorithms: *(1) Graph-Centric Contrastive Learning*, which enables GRENADE to harness intrinsic graph knowledge through relational-aware and augmentation-free contrastive learning; and *(2) Graph-Centric Knowledge Alignment*, which facilitates the exchange and strengthening of knowledge derived from the pre-trained language model encoder and the graph neural network encoder, thereby enhancing the capture of relational information from TAG. We conduct experiments on four benchmark datasets under few-shot and full data node classification, node clustering, and link prediction tasks, and find that GRENADE significantly and consistently outperforms baseline methods.

## 7 Limitations

In this section, we acknowledge the following constraints in our study: *(1) Constraints on the Choice of Backbone Model.* Our choice of the backbone model was restricted to the initialization of "bert-base-uncased" in training GRENADE. This choice was necessitated by the limitations in computational resources available to us. Exploration with alternative PLM backbones such as GPT2(Radford et al., 2019) and RoBERTa(Liu et al., 2019) has not been carried out and represents a promising direction for subsequent studies. *(2) Comparison with Large Language Models.* The natural language processing domain has recently seen breakthroughs with state-of-the-art large language models like LLaMA(Touvron et al., 2023) and ChatGPT (OpenAI, 2023), which have demonstrated exceptional performance in language understanding tasks. Our experiment confirms that GRENADE surpasses existing representation learning techniques in TAG, but the performance of GRENADE relative to these cutting-edge language models remains to be ascertained. *(3) Breadth of Evaluation.* In this work, we evaluate GRENADE primarily through node classification, node clustering, and link prediction tasks. However, there are other relevant evaluation dimensions, such as retrieval, reranking, co-view and others (Cohan et al., 2020). Future work will investigate the applicability and capacity of GRENADE to broader tasks.

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

## A  Dataset Details

We extend ogbn-arxiv and ogbn-products for link prediction and we evaluate models on the test-split from these two node datasets. Specifically, for each source node $i$, we randomly choose its one neighbor as the positive candidate and 1,000 negative candidates and would like the model to rank the positive candidate over the negative candidates. The negative references are randomly-sampled from all the nodes from TAG that are not connected by $i$.

For ogbl-citation2, we also extend it for node classification. Here the task is to predict the subject areas of the subset of the nodes/papers that published in arxiv like ogbn-papers100M. We borrow the labels for ogbl-citation2 from ogbn-papers100M. We align the nodes from ogbl-citation2 and ogbn-papers100M through Microsoft academic graph paper ID. We split the data into training/validation/test by year that the papers published before 2017 is the training dataset, between 2017-2018 is the validation dataset and after 2018 is the test dataset.

## B  Implementation Details

Our proposed methodology was implemented using PyTorch version 1.13.1 (Paszke et al., 2019) and HuggingFace Transformers version 4.24.0 (Wolf et al., 2019). The experiments were executed on A5000, A6000, and RTX 4090 GPUs. For GRENADE, the learning rate is configured at $5e-5$, and AdamW optimizer is employed for training over the course of 3 epochs. The search space for the number of GNN layers, $L$, ranges from $\{1, 2, 3, 4\}$, and further hyperparameter analysis is performed as detailed in Fig. 6. The hyperparameters for few-shot node classification are shown in Tab. 6 and Tab. 7, respectively. It should be noticed that "$-1$" means utilize all the training data for batch_size and all the neighbors for neighbor sampling.

| Hyperparameters | Value |
|---|---|
| hidden_size | 256 |
| dropout | 0.5 |
| lr | 1e-4 |
| epochs | 300 |
| batch_size | -1 |

Table 6: Hyperparameter setting for MLP model in node classification.

## C  Additional Experimental Results

**Full Data Node Classification.**

| Datasets | ogbn-arxiv | ogbn-products | ogbl-citation2 |
|---|---|---|---|
| hidden_size | 256 | 256 | 256 |
| dropout | 0.5 | 0.5 | 0.5 |
| lr | 1e-3 | 1e-3 | 1e-3 |
| epochs | 500 | 50 | 50 |
| batch_size | -1 | 1024 | 1024 |
| neighbors | -1– -1 | 5–10 | 5–10 |

Table 7: Hyperparameter setting for GraphSAGE model in node classification.

**Enhanced Node Classification Model.** Besides MLP and GraphSage, we incorporated the recent Graph Transformer Network, NAGphormer (Chen et al., 2023), into our node classification evaluation. As evidenced by the data in Tab. 8, our proposed approach consistently surpasses the baseline techniques (BERT+MLM, SPECTER, GIANT) in the few-shot and full-data node classification. This performance is consistent with the observations of using MLP and GraphSage.

| Methods | $k=2$ | $k=4$ | $k=8$ | $k=16$ | All |
|---|---|---|---|---|---|
| BERT+MLM | 42.47 | 49.29 | 56.78 | 60.52 | 66.37 |
| SPECTER | 49.89 | 54.79 | 59.66 | 63.26 | 69.11 |
| GIANT | 40.57 | 44.81 | 54.27 | 58.93 | 65.82 |
| GRENADE | **52.73** | **58.28** | **62.53** | **64.97** | **70.85** |

Table 8: ogbn-arxiv node classification result on NAGphormer.

**Additional Ablation Study.** We have the ablation study on ogbn-arxiv and ogbn-products dataset under 7 different model variations. In the second row of Tab. 9, the term ICL refers to instance-wise cross-modality contrastive learning. It indicates that the positive pairs are formed between identically indexed document and node representations, while the negative pairs consist of other document and node representations within the minibatch. From Table 9, have the consistent observation as 5 that each component of GRENADE contributes the performance improvement.

**In-Depth Hyperparameter Analysis.** To evaluate the performance of the graph neural network encoder (GNN), we conduct an analysis of the hyperparameter $L$ as depicted in Fig. 6. Our observations indicate that an optimal performance is achieved when $L = 2$.

**Inference Time Complexity Analysis** When provided with an arbitrary node, GRENADE is capable of generating the textual representation of the node without requiring graph information. This leads

| Methods | ogbn-arxiv | | | | ogbn-products | | | |
|---|---|---|---|---|---|---|---|---|
| | MLP(ACC) | K-means++(ACC) | MRR | Avg. Rank↓ | MLP(ACC) | K-means++(ACC) | MRR | Avg Rank.↓ |
| GRENADE | $73.16_{\pm0.12}$ | $61.96_{\pm0.79}$ | 52.73 | **1.67** | $81.58_{\pm0.18}$ | $73.54_{\pm1.03}$ | 72.39 | **2.00** |
| GC-CL+ICL+NBH-KA | $72.20_{\pm0.20}$ | $60.67_{\pm0.71}$ | 47.74 | 4.33 | $80.90_{\pm0.24}$ | $72.48_{\pm1.42}$ | 65.72 | 5.67 |
| GRENADE w/o GC-CL | $72.83_{\pm0.08}$ | $58.75_{\pm0.99}$ | **55.87** | 3.33 | $80.52_{\pm0.15}$ | $71.25_{\pm0.98}$ | **74.51** | 4.33 |
| GRENADE w/o ND-KA | $72.65_{\pm0.16}$ | $60.69_{\pm0.72}$ | 49.15 | 4.33 | $80.96_{\pm0.29}$ | $73.37_{\pm0.71}$ | 68.47 | 4.33 |
| GRENADE w/o NBH-KA | $73.05_{\pm0.17}$ | $60.50_{\pm0.86}$ | 52.56 | 3.33 | $81.36_{\pm0.30}$ | $72.77_{\pm1.42}$ | 72.08 | 3.33 |
| GRENADE w/o ND-KA+NBH-KA | $72.71_{\pm0.21}$ | $60.34_{\pm1.00}$ | 49.67 | 4.67 | $81.14_{\pm0.17}$ | $73.59_{\pm0.82}$ | 68.71 | 3.00 |
| GRENADE w/o GC-CL+NBH-KA | $72.63_{\pm0.09}$ | $58.66_{\pm0.77}$ | 55.46 | 5.00 | $80.35_{\pm0.20}$ | $71.10_{\pm1.20}$ | 74.27 | 5.33 |
| GRENADE w/o GC-CL+ND-KA | $17.97_{\pm6.48}$ | $22.49_{\pm0.17}$ | 0.55 | 8.00 | $66.76_{\pm0.62}$ | $44.83_{\pm0.26}$ | 21.08 | 8.00 |

Table 9: Ablation study of graph-centric contrastive learning (CL) and knowledge-alignment between PLM encoder and GNN encoder (ND-KA and NBH-KA) on ogbn-arxiv and ogbn-products datasets.

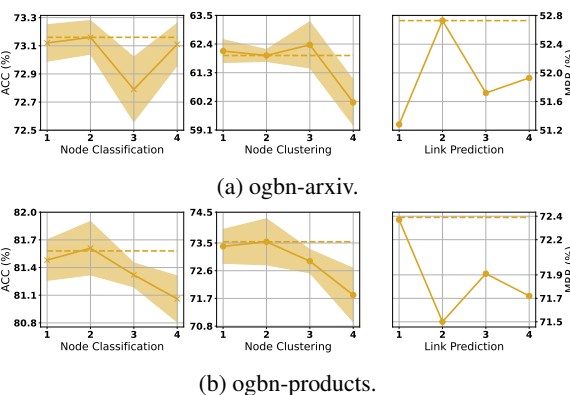

(a) ogbn-arxiv.

(b) ogbn-products.

Figure 6: Hyperparameter analysis on $L$. Dashed horizontal lines are the performance of chosen $l$ in this paper.

to an efficiency that is on par with BERT (Devlin et al., 2018), while achieving approximately a 10% enhancement in performance for full data node classification.

Regarding the training efficiency, though our model requires longer training time compared to text-only PLM/SSL methods like BERT+MLM (~1h24m) and SimCSE (Gao et al., 2021)(around 41m), we are able to achieve a great margin of performance improvement with reasonable additional training time (around 2h50m). Also, our model can achieve stable performance with 3-epoch training, which is more efficient than structure-based contrastive learning method such as SPECTER (Cohan et al., 2020) (around 3h19m for 3 epochs).