# OpenReview forum: "GRENADE: Graph-Centric Language Model for Self-Supervised Representation Learning on Text-Attributed Graphs"
_EMNLP/2023/Conference — EMNLP 2023 Findings_

### Official Review · Reviewer_6SrM · 2023-08-04

**Typos Grammar Style And Presentation Improvements:** None.
**Soundness:** 4

**Excitement:**

4: Strong: This paper deepens the understanding of some phenomenon or lowers the barriers to an existing research direction.

**Missing References:**

None.

**Paper Topic And Main Contributions:**

This paper focuses on self-supervised representation learning on text-attributed graphs. The authors argue that existing methods either struggle to capture the full extent of structural context information or rely on task-specific training labels. To solve such problem, this paper proposes a Graph-Centric Language model named Grenade. Grenade optimizes graph-centric contrastive learning and graph-centric knowledge alignment to exploit the synergistic effect of both pre-trained language model and graph neural network. The experimental results demonstrate the effectiveness of the proposed method.

**Questions For The Authors:**

None.

**Reasons To Accept:**

1. The writing is clear and easy to follow.
2. Two novel self-supervised learning algorithms are proposed to perform contrastive learning and knowledge alignment in a graph-centric way.
3. Extensive experiments show that Grenade achieves significant and consistent improvements over state-of-the-art methods on a wide spectrum of downstream tasks.

**Reasons To Reject:**

I do not see significant weaknesses or omissions in this paper.

**Reproducibility:**

3: Could reproduce the results with some difficulty. The settings of parameters are underspecified or subjectively determined; the training/evaluation data are not widely available.

**Reviewer Confidence:**

2: Willing to defend my evaluation, but it is fairly likely that I missed some details, didn't understand some central points, or can't be sure about the novelty of the work.

---

> ### Author Rebuttal · Authors · 2023-08-29
>
> Thank you for your positive feedback about our work!
>
> Regarding your concern about reproducibility, we would like to mention that all the detailed hyper-parameter settings and their respective analyses are included in the appendix of our paper. This should provide clarity on the configurations used in our experiments. We have also uploaded our source code along with the training scripts to the supplementary materials. These resources are intended to offer other researchers an in-depth view of our implementation, further aiding in reproduction. After the acceptance of our paper, we plan to release the model checkpoints additionally in public.

---

### Official Review · Reviewer_esjL · 2023-08-04

**Soundness:** 3

**Excitement:**

3: Ambivalent: It has merits (e.g., it reports state-of-the-art results, the idea is nice), but there are key weaknesses (e.g., it describes incremental work), and it can significantly benefit from another round of revision. However, I won't object to accepting it if my co-reviewers champion it.

**Paper Topic And Main Contributions:**

The proposed Grenade is a self-supervised graph centric language mode for learning expressive and generalized representation from textual attributed graphs (TAG). The proposed Grenade capture informative textual semantic and graph structure information to addresses the underexplored problem of self-supervised learning on TAG.


**Questions For The Authors:**

A. Why do the authors set the K-hop neighbor as 1? What about 2 or others?
B. What about the effectiveness and generalizability of the proposed method?

**Reasons To Accept:**

1. Clear motivation.
2. Extensive experiments show the Grenade is able to learn efficient representations for node classification and link prediction tasks.




**Reasons To Reject:**

1. Poor presentation of the method section.
2. This paper emphasizes that the proposed method can improve the representation expressiveness. However, it does not prove and analyze the expressiveness.



**Reproducibility:**

4: Could mostly reproduce the results, but there may be some variation because of sample variance or minor variations in their interpretation of the protocol or method.

**Reviewer Confidence:**

4: Quite sure. I tried to check the important points carefully. It's unlikely, though conceivable, that I missed something that should affect my ratings.

---

> ### Author Rebuttal · Authors · 2023-08-29
>
> Thank you so much for providing your feedback! Please find our detailed response addressing your concerns as follows:
>
> > **Q1**. This paper emphasizes that the proposed method can improve the representation expressiveness. However, it does not prove and analyze the expressiveness.
>
> **A1**. Thank you for raising the concern regarding the expressiveness of our proposed method. In our paper, we claim that by eliciting and aligning the knowledge between pretrained language model and graph neural network, our model can learn expressive node representations for each text document. To validate this point, we conducted multiple analyses in our experiments:
> - Quantitative Analysis:
>     1. Few shot node classification: In Table 2, our method outperformed GIANT, SPECTER et al baseline methods by 5% showing and proving the utility of our representations for tasks.
>     2. Node clustering: In Table 3, our representations led to more coherent clusters, evidenced by an improvement of at least 2% in the K-Means++.
>     3. Link prediction: In Figure 2 our method showed a 4% enhancement over the best baseline method SPECTER in ogbn-arxiv, demonstrating the expressiveness in relational patterns.
> - Qualitative Analysis:
>     4. Representation visualization: In Figure 3, we visualized the embeddings using t-SNE, and observed clear distinctions and patterns which further validated the rich expressiveness of our representations.
>
> The consistent superior performance across varied tasks strongly indicates the enhanced expressiveness of the representations generated by our method. We will ensure that the paper emphasizes and details these results more clearly in the revised version by improving the readability.
>
>
> >  **Q2**. Why do the authors set the K-hop neighbor as 1? What about 2 or others? B. What about the effectiveness and generalizability of the proposed method?
>
> **A2**. Thank you for your question. Our choice of setting **K=1** is mainly based on the consideration of simplicity and effectiveness.
> In practice, 1-hop neighbors are good enough for capturing the structure knowledge in our algorithm.
> When we consider 2-hop or 3-hop neighbors, there's an increased potential for introducing noisy information.
> This observation is also consistent with the findings from the related work, SPECTER [1]. We believe our approach balances the trade-off between capturing structural information and mitigating noisy information, leading to stable performance in our evaluation. To further justify our choice, we conducted a hyperparameter analysis by varying the value of $K$ on the ogbn-arxiv datasets. The results, as illustrated in the following table, clearly show that increasing the hop distance degrades performance metrics across node classification (MLP ACC), node clustering (KMeans ACC), and link prediction (MRR).
>
> |     | MLP ACC   | Kmeans ACC | MRR       |
> |-----|-----------|------------|-----------|
> | K=3 | 71.97     | 61.06      | 47.48     |
> | K=2 | 72.21     | 60.60       | 49.59     |
> | K=1 | **73.16** | **61.96**  | **52.73** |
>
>
> [1] Cohan, Arman, et al. "SPECTER: Document-level Representation Learning using Citation-informed Transformers." ACL 2020.
>
> > **Q3**. What about the effectiveness and generalizability of the proposed method?
>
> **A3**. In this paper, we propose a new method for self-supervised representation learning on text attributed graphs.
> Based on our proposed graph-centric contrastive learning and graph-centric knowledge alignment algorithms, we are able to learn a language model that can derive expressive representations for each text node.
> To show the effectiveness and generalizability of our approach, we evaluate the quality of the learned text node representations on different down-stream task, including few-shot node classification, node clustering, and link prediction (more details can be found in our **A1**).
> The superior performance across different downstream tasks clearly show the effectiveness and generalizability of our approach.

---

### Official Review · Reviewer_P3Mr · 2023-08-11

**Soundness:** 3

**Excitement:**

3: Ambivalent: It has merits (e.g., it reports state-of-the-art results, the idea is nice), but there are key weaknesses (e.g., it describes incremental work), and it can significantly benefit from another round of revision. However, I won't object to accepting it if my co-reviewers champion it.

**Paper Topic And Main Contributions:**

The paper is focused on self supervised learning (SSL) on Text augmented graphs by leveraging pre-trained language model (PLM) as well as Graph neural network (GNN).  The proposed method, GRENADE, proposed two new SSL algorithms, namely Graph centric contrastive learning and Graph centric knowledge alignment. Graph centric contrastive learning improves representation of both PLM and GNN encoding via neighborhood information. While Graph centric knowledge alignment enables alignment between the representation of PLM and GNN. The experiments on various graph-specific downstream tasks demonstrate the effectiveness of GRENADE over baselines.

**Questions For The Authors:**

The proposed approach is only adaptable to new tasks on the same graph. Given it demands sufficient compute, it is logical to analyze  how well the proposed framework is adaptable to new graphs of same family (e.g. citation) similar to domain adaptation of LLM. The authors are requested to comment on this aspect of the proposed framework.

For more questions, please refer from WEAKNESS section.

**Reasons To Accept:**

STRENGTHS:
1. Although this work is not the first one to combine PLM and GNN for better representation learning, but it enables language model to infer global structural information resulting to a better Language encoder.
2. Alignment is a much needed module in Multimodal space. The proposed objective of aligning Language and GNN encoder seems to be a very effective for SSL on graph specific tasks as validated by the ablation results.
3. Overall the paper has a novel content and easy to follow.

**Reasons To Reject:**

WEAKNESS:
1. Given that all the experimental datasets are homophilous, it would be interesting to explore the performance of GRENADE in non-homophilous graphs. How much the proposed SSL objectives stand up to in this scenario.
2. There is no discussion on computational effort especially given that method is relying on PLM.
3. There is no comparison with graph-only SSL approaches such as Auto-SSL [1]. Though it will not be an apple to apple comparison since they lack Language encoders but will be interesting to examine the effectiveness of incorporating PLM.
4. In Table 2, it would be nice to include Graph Transformer Networks results in order to compare it with MLP and GraphSage.

[1] Jin, W., Liu, X., Zhao, X., Ma, Y., Shah, N., & Tang, J. (2021). Automated self-supervised learning for graphs. arXiv preprint arXiv:2106.05470.

**Reproducibility:**

4: Could mostly reproduce the results, but there may be some variation because of sample variance or minor variations in their interpretation of the protocol or method.

**Reviewer Confidence:**

4: Quite sure. I tried to check the important points carefully. It's unlikely, though conceivable, that I missed something that should affect my ratings.

---

> ### Author Rebuttal · Authors · 2023-08-29
>
> Thank you for your time and helpful comments. We have addressed the concerns below. If you have any further questions, please let us know.
>
> > **Q1**. Given that all the experimental datasets are homophilous, it would be interesting to explore the performance of GRENADE in non-homophilous graphs. How much the proposed SSL objectives stand up to in this scenario.
>
> **A1**. Thank you for bringing up the consideration of non-homophilous graphs.
> We want to clarify that in this paper we mainly focus on the homophilous graphs, which are more common in the real-world.
> As we mentioned in the paper, our proposed graph-centric contrastive learning and knowledge alignment algorithms also highly rely on the Homophily principle. Handling non-homophilous graphs is definitely an interesting future direction, but it is out of this paper’s scope. But our proposed work has the potential to adapt to non-homophilous graphs with careful design, such as revising the way of selecting positive samples and negative samples in our self-supervised learning objective functions.
>
> > **Q2**. There is no discussion on computational effort especially given that method is relying on PLM.
>
> **A2**. Regarding the training efficiency, though our model requires longer training time compared to text-only PLM/SSL methods like BERT+MLM (~ 1h24m) and SimCSE(~ 41m), we are able to achieve a great margin of performance improvement with reasonable additional training time (~ 2h50m).
> Also, our model can achieve stable performance with 3-epoch training, which is more efficient than structure-based contrastive learning method such as SPECTER (~ 3h19m for 3 epochs).
>
> In terms of the inference time, our method only needs to access the text document, thus it is computationally comaprable with text-only PLM/SSL methods like BERT (around 1h59m for ogbn-arxiv dataset) and more efficient than GNN-based methods. All the above reported time was evaluated on the ogbn-arxiv dataset with an Nvidia A6000 GPU.
>
> > **Q3**. There is no comparison with graph-only SSL approaches such as Auto-SSL [1]. Though it will not be an apple to add comparison since they lack Language encoders but will be interesting to examine the effectiveness of incorporating PLM.
>
> **A3**. Thank you for your insightful feedback.
> It is worth noting that graph-only SSL techniques usually underperform TAG SSL methods (e.g., SPECTER[1] and GIANT[2]).
> This is primarily because Graph-only SSL leverages graph-agnostic methods such as tf-idf, word2vec, BERT to extract numerical features from text and only relies on GNNs to learn the text node representations, which cannot capture the inherent correlations between text attributes and graph structure.
> To comapre with graph-only SSL methods, we have incorporated two Graph-only SSL methods Auto-SSL[3] and VGAE[4] in the node classification evaluation (with full training data, the baseline results are adopted from the corresponding paper) on ogbn-arxiv. As observed from the results in the subsequent table, Text+Graph SSL outperforms Graph-only SSL in performance metrics and our model GRENADE achieves the best performance.
>
>
> |    SSL Types            | Method            | ogbn-arxiv |
> |----------------|-------------------|------------|
> | Graph-Only SSL | Word2Vec+Auto-SSL | 69.13      |
> | Graph-Only SSL | Word2Vec+VGAE     | 72.04      |
> | Graph-Only SSL | BERT+VGAE         | 72.44      |
> | TAG SSL        | GIANT             | 74.59      |
> | TAG SSL        | GRENADE           | **75.00**  |
>
> [1] Cohan, Arman, et al. "SPECTER: Document-level Representation Learning using Citation-informed Transformers." ACL 2020.
>
> [2] Chien, Eli, et al. "Node Feature Extraction by Self-Supervised Multi-scale Neighborhood Prediction." ICLR 2021.
>
>
> [3] Jin, Wei, et al. "Automated Self-Supervised Learning for Graphs." ICLR 2021.
>
>
> [4] Kipf, Thomas N. et al. "Variational Graph Auto-Encoders."  NIPS 2016.
>
> > **Q4**. In Table 2, it would be nice to include Graph Transformer Networks results in order to compare it with MLP and GraphSage.
>
> **A4**. Thank you for your suggestion. In addition to MLP and GraphSage, we have added a recent Graph Transfomer Network NAGphormer [1] as the classification model in the few-shot node classification evaluation. According to the results shown in the following table, our proposed method consistently outperforms the baseline methods (BERT+MLM, SPECTER, GIANT) on the few-shot node classification task, which is aligned with the observations of using MLP and GraphSage.
>
> **ogbn-arxiv node classification result on NAGphormer**
>
> |   Methods       | k=2       | k=4       | k=8       | k=16      | all       |
> |----------|-----------|-----------|-----------|-----------|-----------|
> | BERT+MLM | 42.47     | 49.29     | 56.78     | 60.52     | 66.37     |
> | SPECTER  | 49.89     | 54.79     | 59.66     | 63.26     | 69.11     |
> | GIANT    | 40.57     | 44.81     | 54.27     | 58.93     | 65.82     |
> | GRENADE  | **52.73** | **58.28** | **62.53** | **64.97** | **70.85** |
>
> [1] Chen, Jinsong, et al. "NAGphormer: A tokenized graph transformer for node classification in large graphs." ICLR 2022.
>
> > **Q5**.  The proposed approach is only adaptable to new tasks on the same graph. Given it demands sufficient compute, it is logical to analyze how well the proposed framework is adaptable to new graphs of same family (e.g. citation) similar to domain adaptation of LLM. The authors are requested to comment on this aspect of the proposed framework.
>
> **A5**. Thank you for highlighting the importance of adaptability of our proposed approach to new graphs. Based on your suggestion, we conducted a new set of experiments to assess the adaptability of our model across graphs of same family as well as different families. Specifically, our model was trained on ogbn-arxiv and subsequently evaluated on ogbl-citation2 (same family) and ogbn-products (different families). As reported in the below table, our model achieves better adaptability compared to  the baseline methods in both cases.
>
> |          | arxiv->citation2 |                  |                  |  arxiv->products |                  |          |
> |----------|:----------------:|:----------------:|:----------------:|:----------------:|:----------------:|:--------:|
> |          | k=16 ACC      | KMeans ACC       | MRR              | k=16 ACC      | KMeans ACC       | MRR      |
> | BERT+MLM |        30.9      |        25.78     |        9.04      |        44.75     | 45.74            | 27.22    |
> | SPECTER  |        33.01     |        36.71     |        23.11     |        47.52     |        58.85     | 45.39    |
> | GIANT    |        34.84     |        39.33     |        18.61     |        45.56     | 57.31            | 45.48    |
> | GRENADE  | **35.65** | **42.68** | **27.17** | **47.56** | **60.14** | **52.50** |

---

### Meta-Review · Area_Chair_Sq16 · 2023-09-19

**Recommendation:** 3

**Metareview:**

The paper introduces GRENADE, a self-supervised learning (SSL) approach for text-augmented graphs that leverages both pre-trained language models (PLMs) and graph neural networks (GNNs). GRENADE proposes two novel SSL algorithms, Graph-centric Contrastive Learning and Graph-centric Knowledge Alignment. The former improves the representation of PLM and GNN encoding by incorporating neighborhood information, while the latter aligns the representations of PLM and GNN. The experiments conducted on various graph-specific downstream tasks demonstrate the effectiveness of GRENADE over baselines.

The strengths of the paper lie in its novel content and clear presentation. GRENADE introduces innovative SSL objectives that effectively leverage both PLM and GNN representations. The alignment module addresses a crucial need in multimodal learning, and the ablation results validate its effectiveness. The paper is easy to follow, making the proposed method accessible to the reader. Additionally, the extensive experiments provide substantial evidence of GRENADE's performance superiority over existing methods.

While the paper presents several strengths, there are some areas that could be improved. It would be valuable to explore GRENADE's performance on non-homophilous graphs to assess the robustness of the proposed SSL objectives in diverse scenarios. The computational effort required by GRENADE is not discussed, which is an important consideration, particularly when relying on PLMs. Comparisons with graph-only SSL approaches like Auto-SSL could provide insights into the effectiveness of incorporating PLMs. Additionally, including results from Graph Transformer Networks in Table 2 would enable a comparison with other graph-based SSL methods like MLP and GraphSage.

---

### Decision · Program_Chairs · 2023-10-07

**Decision:**

Accept-Findings

**Comment:**

The paper introduces GRENADE, a self-supervised learning (SSL) approach for text-augmented graphs that leverages both pre-trained language models (PLMs) and graph neural networks (GNNs). GRENADE proposes two novel SSL algorithms, Graph-centric Contrastive Learning and Graph-centric Knowledge Alignment. The former improves the representation of PLM and GNN encoding by incorporating neighborhood information, while the latter aligns the representations of PLM and GNN. The experiments conducted on various graph-specific downstream tasks demonstrate the effectiveness of GRENADE over baselines.

The strengths of the paper lie in its novel content and clear presentation. GRENADE introduces innovative SSL objectives that effectively leverage both PLM and GNN representations. The alignment module addresses a crucial need in multimodal learning, and the ablation results validate its effectiveness. The paper is easy to follow, making the proposed method accessible to the reader. Additionally, the extensive experiments provide substantial evidence of GRENADE's performance superiority over existing methods.

While the paper presents several strengths, there are some areas that could be improved. It would be valuable to explore GRENADE's performance on non-homophilous graphs to assess the robustness of the proposed SSL objectives in diverse scenarios. The computational effort required by GRENADE is not discussed, which is an important consideration, particularly when relying on PLMs. Comparisons with graph-only SSL approaches like Auto-SSL could provide insights into the effectiveness of incorporating PLMs. Additionally, including results from Graph Transformer Networks in Table 2 would enable a comparison with other graph-based SSL methods like MLP and GraphSage.